# Studying Collaboration Dynamics in Physical Learning Spaces: Considering the Temporal Perspective through Epistemic Network Analysis

**DOI:** 10.3390/s21092898

**Published:** 2021-04-21

**Authors:** Milica Vujovic, Ishari Amarasinghe, Davinia Hernández-Leo

**Affiliations:** Department for Information and Communication Technologies, Universitat Pompeu Fabra, 08018 Barcelona, Spain; ishari.amarasinghe@upf.edu (I.A.); davinia.hernandez-leo@upf.edu (D.H.-L.)

**Keywords:** learning space, collaborative learning, table shape, group size, gender, epistemic network analysis

## Abstract

The role of the learning space is especially relevant in the application of active pedagogies, for example those involving collaborative activities. However, there is limited evidence informing learning design on the potential effects of collaborative learning spaces. In particular, there is a lack of studies generating evidence derived from temporal analyses of the influence of learning spaces on the collaborative learning process. The temporal analysis perspective has been shown to be essential in the analysis of collaboration processes, as it reveals the relationships between students’ actions. The aim of this study is to explore the potential of a temporal perspective to broaden understanding of the effects of table shape on collaboration when different group sizes and genders are considered. On-task actions such as explanation, discussion, non-verbal interaction, and interaction with physical artefacts were observed while students were engaged in engineering design tasks. Results suggest that table shape influences student behaviour when taking into account different group sizes and different genders.

## 1. Introduction

In the field of education, there is ongoing discussion about the meaningful effect that learning spaces seem to have in facilitating and supporting learning scenarios and as a relevant element of the learning design [1,2]. In particular, the collaborative learning approach to pedagogy—as opposed to traditional lectures—has introduced versatile dynamics into the interaction between students, peers, and teachers, but also with the environment [3,4]. However, how learning spaces support or inhibit the potential of these dynamics has not been sufficiently explored. By understanding the effect of elements of the environment on student behaviour in the collaborative process, significant contributions can be made to inform design for productive collaborative learning. 

Indeed, transforming traditional classrooms into spaces that support active learning models, where collaboration plays a major role, requires adapting the space to them [5,6]. Collaborative spaces, unlike traditional classrooms, feature elements that should support the actions characteristic of collaboration. Therefore, through the examination of these actions and the way in which they are represented in the physical domain, insight may be gained into the relationship students have with the space surrounding them and that directly facilitates collaboration. However, learning analytics methods focused on studying collaboration rarely include spatial aspects as factors influencing collaboration. Although mostly static, space plays a role in the development of collaborative activities by providing a framework within which actions occur. Furthermore, aspects that have been shown to influence collaboration such as group size and gender should be explored in relation to the learning space. The differences between dyads and triads and the potential superiority of one group size over another, is a research topic that has been present for a long time [7,8,9]. Gender is also a factor influencing the dynamics in collaboration, primarily due to gender imbalance which is extensively researched [10,11,12,13,14,15]. The influence of these factors on the dynamics of collaboration has been reported in the literature as well as the influence of learning space. However, the interaction of these factors requires more detailed research. As data science and methods become more sophisticated, new opportunities for exploring this issue are emerging. The investigation of collaborative actions, their evolution over the duration of a learning activity, and the potential impact of the environment on this dynamic require methods of analysis that include the development of actions over time and their interconnections. It is insufficient to focus on coding and counting of actions without monitoring and understanding their interconnectedness [16] through the analysis of the temporal perspective of collaborative activity [17,18].

A previous study [19] examining the influence of table shape on the behaviour of elementary school and university students found significant differences. The results suggested that elementary school students participate in collaborative activity more, when using round tables. However, the differences between different table shapes were not significant for university students. This study focused on analysing the influence of different table shapes on student behaviour during collaborative activity, but did not consider the temporal dimension of the actions. Yet a temporal perspective is crucial to understanding the development of collaborative processes as they may reveal interactional patterns [20], show how collaborative actions can encourage socially shared planning and regulation [21], and provide more detailed insight into the active learning processes of groups [22].

Temporal perspective analysis has been adopted in a number of studies, in which various techniques such as temporal pattern analysis [23], variable- and event-centred analysis [22], sequential analysis [24], sequence and process mining [25,26], and dynamic multilevel analysis [27] were applied. Therefore, the behaviour of university students was re-examined from a new perspective and motivated by studies that indicate different possibilities for temporal perspective analysis. More specifically, the focus was on the new techniques that enable the parallel analysis of several variables such as table shape, group size and gender. With the ability to model and analyse multiple conditions in parallel, as well as to examine the frequency of co-occurrences of actions, epistemic network analysis (ENA) is suitable for studying temporal aspects of the collaboration process in observed context [28,29].

The aim of this study is to investigate to what extent a temporal perspective in the analysis of student behaviours can improve our understanding of the potential influences of learning space elements in collaborative learning activities. By adding another explanatory factor, the temporal perspective, this study could contribute with clarification of previous findings on these differences and how the characteristics of learning space play a role in students behaviour. Therefore, we use ENA for the analysis of temporal correlations between students’ on-task actions. The aim is not to label one space as better or worse, but rather to investigate whether an analysis of temporal dynamics in collaboration can provide evidence of the potential influences and thus facilitate informed decision making in learning space design that is aligned with a specific pedagogical intent and its related learning design facets.

To achieve this aim, this paper focuses on table shape (round vs. rectangular tables) as a relevant element in the design of learning spaces and on group size as a key design facet in collaborative learning. In addition, the paper also explores the gender perspective in a study of the effects of different table shapes.

(RQ1) What can a temporal analytics perspective tell us about the effect of table shape on student behaviour in different group sizes (dyads and triads) during a collaborative activity?

(RQ2) What can a temporal analytics perspective tell us about the effect of table shape on the behaviour of different genders (female and male) during a collaborative activity?

Therefore, this paper presents an analysis of the impact of table shape on student behaviour, with a focus on the temporal component of collaboration. Using ENA as a tool that incorporates the temporal component into the analysis of coded actions by modeling co-occurrences of action bypasses the classical approach of coding and counting and provides a deeper understanding of collaborative development over time. In comparing two conditions—round and rectangular tables, and their interaction with group size and student gender—the aim is to identify potential differences. The second section surveys the literature relevant to this study, which includes indicators of collaboration, learning space, group size and gender in interaction with learning space, temporality in the analysis, and ENA. The third section outlines the research aims and questions while the next covers the methods used in applying ENA in this study. The fifth section presents the findings of the analysis, while the sixth section discusses the results. Finally, the seventh and eighth sections consider the limitations and conclusions of the study as well as future work.

## 2. Background

In order to ground the analysis of the learning space in which we examine the student behaviour during collaborative learning activities, it was necessary to blend several different domains that converge in the field of learning analytics. This study referred to previous work on the indicators of high-quality collaborative learning, impact of learning space on behaviour, temporality as the focus of analysis, and finally, ENA. 

### 2.1. Indicators of Fruitful Collaborative Learning (in New Learning Spaces)

Indicators of productive collaboration can be found in the actions that students perform during collaborative learning activities. Those actions may be categorised into two groups: on-task and off-task. Students engaging in actions or interactions that are unrelated to the task is considered to be off-task behaviour [30], whereas paying attention during instructions or focusing on group or individual work is on-task behaviour [31]. Furthermore, on-task behaviour is defined as attending to assigned tasks, focusing on the appropriate materials, manipulating learning objects, and maintaining eye contact with the teacher, team members, or task objects [31]. In collaborative learning, these actions are essential for problem solving [32] and for generating the social awareness and overall positive perception of group members’ interdependence and accountability that underpin fruitful collaboration [33]. Furthermore, it has been reported [34] that when conducting collaborative work in classroom settings, groups of students who exhibited on-task actions more often generated better solutions to the problems.

To define the analysis of on-task actions, the classification of activities based on their characteristics plays a major role. Two main characteristics of on-task actions may be distinguished: active and passive [35,36]. The literature indicates the actions common to collaborative learning and whose analysis is used in this study to better understand learning space effects. The action of *explaining* represents a passive action and refers to reading out loud or talking with the objective of clarifying instructions or ideas. This is considered beneficial as it reorganises the material in a new way, develops new perspectives, and resolves inconsistencies, which would not be accomplished as comprehensively when done individually [24,36]. Additionally, *discussion* is considered an active social on-task action that engages more than one person in dynamic interaction [35] which is beneficial for collaboration. Another on-task action in the social category is non-verbal participation, meaning actions and gestures of listening and observing without extended verbal engagement, which has been shown to be highly significant in collaboration. Webb et al. [37] demonstrated that, in certain cases, participants contribute more when they are not under pressure to say something. In physical on-task actions, interaction with the artefacts is essential for completing the collaborative activity task. Artefact use indicates engagement with the task and, when it occurs during explanations or discussion, reinforces collaboration and provides more balanced participation [38]. Overall, the analysis of on-task actions plays a key role in analysing collaboration, and indirectly in the analysis of the space in which collaboration takes place. Grounding the analysis in previously presented findings in the literature that reveal the indicators of good collaboration, this study uses defined on-task actions to assess student behaviour and analyse learning space effects.

### 2.2. Learning Spaces

Although progress in the development of in-person learning models has been evident over recent decades, new learning spaces often neglect the pedagogical vision [5]. With this in mind, the authors [5] stress out that there is a clear need for an integrated design in order to achieve a balance between learning models and the physical environment in which they are implemented. Besides their subtle influence on students, learning spaces possess the power to encourage or inhibit teachers as they create their learning design. Rogers et al. [39] presented a study on the learning space preferences of higher education students, in which the general consensus among the students was that learning spaces indeed affect the outcome of learning activities. 

Furthermore, Beckers et al. [40] found that the informal arrangement of spatial elements encourages more collaboration. Moreover, studies on the physical characteristics of learning space such as colours, light, space shape, and table shape report differences in student behaviour when different conditions are applied [41,42,43]. More specifically, tables with curved, organic shapes have been found to reinforce more on-task student behaviour in active learning classroom systems [44]. The application of round tables in active learning classrooms has been shown to encourage active discussion based on group activities [45]. However, a number of students that used informal learning spaces reported that they did not feel comfortable because they did not have their own familiar space to overcome certain difficulties encountered. Carvalho and Yeoman [6] argue that research on learning spaces requires a more contextualised and less generic understanding of tool and space properties and how they can influence learners and their actions.

Newly-configured learning spaces that are becoming more prevalent in practice incorporate recently developed teaching practices and emerging technologies that are dedicated to team-teaching and collaboration between students [46]. These collaborative environments provide opportunities for more interaction and stimulate innovation due to shared reflections and inquiries, which result in robust and constantly developing collaborative practice. However, with all the innovation introduced into classrooms, more research is needed that can corroborate or challenge the benefits of learning space design. In order to do so, this study focuses on two common influential factors on collaboration, group size and gender, that possibly moderate the effects of learning space on collaborative learning. Selection of these factors will be further elaborated upon in the following paragraph.

### 2.3. Group Size and Gender as Moderators of the Effects of Learning Spaces on Collaborative Learning

Group size and gender have been present in the research of collaborative learning for quite some time as moderators of collaboration. When considering the different group sizes proposed for collaborative learning activities, opinion is divided as to whether dyads or triads develop better collaborative strategies. Carvalho et al. [7] point out the benefits of dyads in terms of the possibilities for students to observe each other and exchange ideas and strategies to improve common performance. Another reported benefit of dyads is more optimised use of equipment among two students, which leads to the efficient completion of practical assignments [8]. 

However, when dealing with collaborative learning, research suggests that dyads should be considered as peer learning, while triads involve real collaboration [47]. In this sense, triads are more likely to foster complex behaviours such as coalitions, negotiations, and conflict, which are all beneficial for learning. A study examining pre-service teachers during teacher preparation programmes found evidence of complexity when working in triads such as benefits from learning from each other, and in support and comprehensive feedback about work being done, as well as limitations in terms of concerns about dependency, loss of individuality, and increased competitiveness [3]. Other benefits that triads have in comparison to dyads are reported [9] to include prompting novel perspectives and enhancing problem-solving abilities, though among the disadvantages are conflicts within group Despite the latter, however, triads are more likely to develop various perspectives within collaborative tasks.

When considering another common moderator of collaboration, gender, previous research has yielded interesting findings. Gender differences have been studied in various educational approaches such as formal learning, informal learning, game-based learning, collaborative learning. Differences between male and female subjects appear to be existing throughout these approaches and at various ages. A study examining gender differences in the achievement of kindergarten children in robotics and programming had the aim to establish if boys and girls at this age have equal successful rates [10]. The young age of children chosen for the study was used to nullify the gender stereotypes that tend to surface in later years. Finally, no significant gender differences were found and the study underlines that further research and comparison of kindergarten children with older children is needed to better understand if reducing stereotype threat at young age can reduce gender differences in older age. 

When observing usage of tangible interfaces in collaborative activity, Sapounidis, et al. [11] reported how this kind of technology applied to collaborative learning can prevent formation of gender stereotypes in early age. In their other study, Sapounidis et al. [12] continued research on children’s preferences related to the design of programming interfaces and its relation to gender and age. Limited research implied the existence of gender effects, where male participants preferred graphical user interfaces, while female participants preferred tangible user interfaces. Another study examined how a significant growth in access to computers in school and at home caused gender and age differences in the computer attitudes among British secondary school students [13] (Colley and Comber, 2003). Findings report on the existence of the gender gap, where among younger and older students, older girls were less favorable towards technology.

Furthermore, a study examining women’s lower interest in video games compared to men, conducted in Germany, shows an association between gender and personality traits [14]. Namely, women described themselves as less competitive and showed a lack of self-confidence when it came to overcoming competitive situations in games. Interesting research on the collaboration between older men and women, has shown that interpersonal factors related to gender may have considerable influence on the experience of collaboration than the task itself [15]. The least positive situation perceived by women participants was when they were paired with an unfamiliar male partner. The reason they listed was the competitiveness in the interaction.

Moreover, the gender composition of groups tends to be a relevant factor in collaborative learning processes. Wiley and Jensen [48] report that groups which are heterogeneous in terms of both gender and skills benefit more from collaborative learning than those that are homogenous. Later research from Cen et al. [49] extended study to female-only groups and provided evidence that certain forms of gender distribution are more conducive to collaborative learning, with female-only and balanced-gender grouping shown to be the most conducive. Existing evidence also suggests gender may have implications for individual student behaviour when collaborating. For example, in a study by Zhan et al. [50], female students claimed that they employed more collaborative learning strategies than their male peers. Further, Stump et al. [51] found that female students sought help from other students with higher frequency in collaborative activities, even if this made them feel like a group member with less knowledge. Another study examined how students’ individual learning performances and knowledge elaboration processes in computer-supported collaborative learning (CSCL) differed between dyads with different gender distributions [52]. This study found that female-only dyad participants outperformed female peers in mixed-gender dyads, while this was not the case with male-only dyads. Finally, there are studies that present issues related to female students experiencing bias, as in engineering programmes where they were disrespected by the male students [53].

Given the need to explore the impact of learning space in more detail and taking group size and gender as factors through which it may be examined, finding a method that would provide useful results in such a complex context was necessary. A temporal analysis perspective offers an interesting opportunity to examine the development of actions in different conditions over time and the differences that occur in them when different group sizes and genders are considered. Temporality as an element of analysis will be discussed further in the following section.

### 2.4. Temporality as the Element of Analysis

The importance of the temporal perspective in learning, as a developmental process, has long been established [54,55,56]. However, there is insufficient use of temporal information from learning data and insufficient exploration of temporal concepts [17]. Only recently have the identification, measurement, and analysis of temporal features of learning attracted the close attention of researchers [9]. This attention has focused on various aspects of temporal analysis in the context of learning such as temporal data types, temporal data visualisation, and analytical methods, as well as their practical application [18]. 

Numerous studies have investigated the development of collaborative learning activity over time and the importance of the order of events in order to better understand the process of collaboration. Reimann [22] points out the relevance of time and order in active learning processes, especially in particularly problematic contexts such as group work. In addressing this issue, when considering the types and processes in regulated collaborative learning, Malmberg et al. [21] report that temporal analysis is useful in showing how collaborative actions related to task execution encouraged socially shared planning and regulation. Interesting conclusions were drawn in a study that looked at individual member contributions in a group discussion [20], in which the temporal evolution of interactional patterns revealed the importance of the first phase of the collaborative process on the overall outcome. Furthermore, Molenaar [57] emphasizes the relevance but also the challenges that arise when attempting to forward temporal analysis as part of learning analytics, such as the multidimensionality of time, different analysis techniques, segmentation of time, differences between micro and macro levels, and the need for confirmatory studies. Moreover, temporal analysis requires specific analysis techniques that can provide deeper insight into the connections between actions as they take place over time. To this end, ENA facilitates the modelling of collaborative interactions while focusing on the temporal order and the co-occurrences of events. 

### 2.5. Epistemic Network Analysis

Quantitative ethnography (QE) is an approach that merges quantitative and qualitative approaches to uncover meaningful patterns in data [58]. The large-scale data generation of digital learning environments today creates opportunities to apply QE to gain meaningful insights into learning and teaching processes [51]. ENA is a statistical tool that exemplifies QE and aids in modelling connections among elements in qualitatively coded datasets [59,60]. 

ENA generates dynamic network models using discourse data through several steps. First, for a given unit of analysis (which could be a collaborative group, a concept, etc.), ENA accumulates the co-occurrences of codes within a defined conversation. This results in the creation of dynamic network models that visualise the structure of connections between coded elements in discourse [59]. Dynamic network models generated using ENA consist of nodes and edges [59]. The nodes represent the codes in discourse data and the network edges connect nodes in the model. The thickness of these network edges represents the relative frequency of the co-occurrences between two codes. Therefore, a thick edge represents a strong connection between nodes and a thinner edge represents a weaker connection.

In the domain of learning analytics, ENA is a tool that provides exceptional opportunities due to its ability to quantify qualitative data and provide an overview of the entire process in terms of the connectivity of its data over a period of time [59]. Current work in this field demonstrates the diversity of the application of ENA and its benefits when it comes to quantitative and qualitative data [28,61]. For instance, ENA has been applied to understanding students’ critical thinking, participation in games, mentoring, and teaching processes. Recent empirical studies have also shown that ENA, in concert with other data analytics techniques such as social network analysis [62] and process and sequence mining [63], can complement each other. The combined methods can provide a complete ontological viewpoint into diverse learning processes such as self-regulated learning and collaborative learning [62,63].

In the context of CSCL, ENA has been applied for different modelling purposes. For instance, Shum et al. [60] have proposed a multimodal matrix inspired by the concepts of QE, producing guidelines on how information presented in a multimodal matrix can be used to deliver feedback to co-located collaborative teams in the context of nursing education. The detection of differences between the connections made by students with high learning gains versus those with low learning gains during collaboration was also the subject of a study that applied ENA [64]. Additionally, in the context of collaborative learning, multiple studies have employed ENA to analyse participants’ behaviour. Andrist et al. [28] used ENA to examine how networks of shared gaze in dyads evolve over longer time windows. Furthermore, Swiecki and Shaffer [65] applied ENA in order to examine cognitive and social patterns in collaborative problem-solving processes of military teams in training. Csanadi et al. [16] examined the temporality of verbal data and compared traditional coding-and counting methods with ENA. Their findings show that ENA has great potential in research of temporality in verbal process data. Application of ENA when employing games to improve collaborative learning skills, was used for the analysis of the collaborative discourse of teams of students [66]. The focus was on determining connections between language style, scientific practice and communication responses.

## 3. Method

### 3.1. Research Setting and Participants

The experimental setup for authentic collaborative learning activity was organised in a motion capture laboratory, where students were invited to participate in an extracurricular activity. In the recruiting process, from more than 150 volunteers who expressed interest in the training, 36 university students with no prior knowledge of the topic were selected from different engineering degrees and different years, with an equal number of male and female participants. These 36 students (aged 18–24) formed 12 jigsaw groups, from which we analysed the data of 8 groups comprising 24 subjects. The subject selection criteria for analysis was: good camera coverage in order to obtain valid data, balance of gender as much as possible, and balance of table shapes. Of the 24 subjects selected for the data analysis, 12 subjects used rectangular tables and 12 round ones (Figure 1). Within the analysed dataset of 24 subjects, there were 11 female and 13 male participants. All groups were mixed-gender groups and, due to the odd number of members in the home groups (triads), the distribution varied between groups of: (a) two female students and one male; and (b) two male students and one female. 

### 3.2. Materials and Task Description

Students participated in a collaborative problem-solving activity in which specific physical computing artefacts were to be designed. Following a jigsaw method [67], each session started with two groups of three members each. A jigsaw pattern includes alternation between two group sizes during a collaborative activity with the aim to help students gain an expertise and improve their home group work by applying that expertise. Therefore, after being given instructions for a divisible task, they were organised into three different expert groups of two members (each coming from different initial groups) for a second phase of the activity in which each group worked on a sub-task. The jigsaw pattern further instructed that after finishing the sub-task, students returned to their home group and continued work on the overall task. Triads and dyads were supported with laptops, although in some cases dyads that were assigned a “design expert” role decided to remove laptops from the table when they did not need them. The task was open-ended, which meant that each group could produce a different design. At the end of the activity, each group presented their work. The activity lasted 90 min, required no prior knowledge, and was designed in a way so that each group member had close to the same workload when conducting her/his part of the task. This organisation of the experiment made it possible to conduct an analysis of the interaction between table shapes, group size, and gender, as presented in Figure 2. 

The participants were asked to design, programme, and build an interactive toy that was to be designed using electronics connected to an open-source electronics platform and additional elements such as cardboard and paper. The difficulty level was adapted to the student profiles (who had no experience with this specific electronic platform) and they were provided with the information necessary for each step of the process. Students were informed of the data collection and analysis that followed this experiment, which was approved by an ethics committee. Informed consent was obtained from the students before the experiment.

### 3.3. Data Collection and Analysis

Data collection was performed with two video cameras used to record the experiments. They were positioned to cover the activity from different angles and at a height of two meters so as to avoid occluding student actions as much as possible. All sessions of the experiment had the same lighting, room temperature, surrounding furniture (except tables used for the activity), researchers present, and sounds in the laboratory (caused by the air conditioning system).

The analysis was based on coding of student actions, where the coding system of on-task activities was established by overlapping information from literature used in similar scenarios and observations of participant behaviour [48,49,50]. All actions that were not classified as on-task were recorded under the common category ‘off-task actions’. Codes included explanation, discussion, and non-verbal interaction as social on-task actions, and interaction with physical artefacts as physical on-task action. Also, the code ‘off-task action’ was included together with the other codes (Table 1). Table 2 presents examples of the defined codes and how they were segmented (when they began and ended) in the specific context that was the subject of the study. Inter-rater reliability was established (values for percentages and Cohen’s kappa were greater than 81.5% and 0.626, respectively). After data were collected and coded, ENA was used to model connections in coded data and to represent them using dynamic network models. We chose the moving stanza window method to select the stanzas within which the connection accumulation was required to be modelled. In other words, the stanza window represents a segment with a certain number of codes within which we want to observe the connection. We selected a moving stanza window size of 3. In this case, each code was observed in relation to the adjacent two codes. Since the stanza was shifted by one code and included two adjacent ones for the observed context where the actions followed one another, this approach was informative enough. 

## 4. Results

ENA was used to model student behaviour in two different learning environments, which were defined by the shape of the table that students used during the collaborative activity. Two group sizes (dyad and triad) as well as two genders (female and male) were observed in both conditions. Student activities were coded and epistemic networks were generated for each of the cases analysed. In order to examine the differences between conditions, a difference network was generated by subtracting the average connection strengths for actions in each condition. The sections below present the results for the four cases analysed, with two considering the effects of table shape on different group sizes and two considering the effects of table shape on different genders. 

In each of the cases analysed and presented, the networks nodes represent each coded action, while the edge weights represent the relative amount of mutual co-occurrence of each of the actions. The network centroids can be described as something similar to the centre of the mass of an object [29] and each condition has its own centroid in this case. To be more specific, the centroid is observed in the context of the projection space and represents the arithmetic mean of all edge weights for the observed network model in that space [28,29]. In this way, the centroid summarises the whole network. With multiple centroids in the ENA model, differences between networks of different conditions are made visible.

### 4.1. Effect of Table Shape on Different Group Sizes

Figure 3 presents the difference networks of co-occurrences of actions for triads in two different conditions (rectangular and round tables). The network models the correlation structure of the five listed actions cumulatively for all triads that participated under each of the conditions. The networks of round tables are presented in blue, while those of rectangular tables are in red. The centroids for round and rectangular table conditions are located at different positions on the *x*-axis, indicating differences in the arithmetic mean of the edge weights for both conditions. This denotes an overall difference in edge weights suggesting that the most frequent co-occurrences between actions under these two conditions are not the same. The strength of the connections between the actions in the case of triads is different between round and rectangular tables. It should be noted that, for round tables, stronger edges exist between interaction with physical artefacts (IPA) and discussion (Ds). This implies that students took turns performing these two actions more often than taking turns with other actions. The proximity of the centroid also confirms an overall prevalence of action co-occurrences under this condition, favouring the alternation of two actions (interaction with physical artifacts (IPA) and discussion (Ds)) that are positioned on the far edge of the projection space. 

The position of the centroid for rectangular tables is closer to the centre of the projection space than that for round tables, which indicates less pronounced co-occurrences of a certain pair of actions. However, difference networks show how certain co-occurrences of actions are more present than others. For rectangular tables, the most frequent co-occurrence was between the action of non-verbal interaction (Nv) and discussion (Ds) (see Figure 3). This shows that students tend to be more engaged in alternating between those two actions than between any other actions when rectangular tables are used. Furthermore, as shown in Figure 3, off-task action (off) is more connected to other actions when rectangular tables are used, which suggests this event is more common under this condition. Under both conditions, discussion (Ds) is the most common co-occurring action. With round tables, discussion alternates with interaction with physical artifacts (IPA), while with rectangular ones it alternates with non-verbal interaction (Nv).

In the case of dyads (Figure 4), the network centroids under the two conditions studied, as with triads, are located in different places in the projection space. The centroid for round tables, shown in blue, is located closer to the border of the projection space, which is defined by the actions of interaction with physical artefacts (IPA) and non-verbal interaction (Nv). This indicates that these co-occurrences between these two actions are more common when compared to other conditions. The centroid for rectangular tables, shown in red, is located closer to the opposite border of the projection space, which is defined by the discussion action, demonstrating that, in the case of rectangular tables, discussion is the action that occurs most frequently. When it comes to the co-occurrences of specific actions, co-occurrences between non-verbal interaction (Nv) and interaction with physical artefacts (IPA) are higher with round tables. This suggests that when using round tables, students communicate less with each other if they are engaged in working with artefacts than is the case with rectangular tables. On the other hand, with rectangular tables, co-occurrences between interaction with physical artefacts (IPA), explanation (Ex), and discussion (Ds) are more frequent than with round tables. This indicates more interpersonal verbal communication (both discussion and explanation) while working with artefacts when dyads use rectangular tables. 

### 4.2. Effect of Table Shape on Gender

Further analysis focused on the influence of table shape on the behaviour of students of different genders when working in groups of two or three members. Figure 4 shows the ENA network for female students’ actions under both conditions. The differing position of the centroids evinces the difference between the two conditions. The centroid and co-occurrences of actions that are more frequent with round tables is shown in blue. The centroid is located closer to the border of the projection space formed by non-verbal interaction (Nv) and interaction with physical artefacts (IPA), meaning that these actions co-occur more often under this condition. This is confirmed by the pronounced blue line representing the edge between these two actions. On the other hand, the centroid for rectangular tables (in red) is located closer to the border of the projection space defined by non-verbal interaction (Nv), discussion (Ds), and off-task actions (off). Together with a pronounced red line, the network suggests that co-occurrences between non-verbal interaction (Nv) and off-task actions (off) are more frequent with rectangular tables. Furthermore, as shown in Figure 5, other nodes in the network are connected to the ‘off’ node, indicating that female students are more engaged with off-task actions when they use rectangular tables.

In the case of male students, as in the previous cases, the positions of the centroids for both conditions also indicate differences between the conditions (Figure 6). The round table centroid, shown in blue, is located close to the node that represents interaction with physical artefacts (IPA). Furthermore, when observing the weight of the edge between interaction with physical artefacts (IPA) and discussion (Ds), it is evident that male students are more frequently engaged in alternations between these actions with round tables than with rectangular ones. Also, co-occurrences between non-verbal interaction (Nv) and discussion (Ds) are more prevalent when male students use round tables. The centroid for rectangular tables, located close to the border of the projection space defined by non-verbal interaction (Nv) and discussion (Ds), together with the pronounced edge weight between these two nodes, shows that these actions co-occur more frequently when rectangular tables are used.

## 5. Discussion

The overall aim of this study was to evaluate whether the adoption of the temporal perspective in the analysis offers insights into how different table shapes affect collaboration with different group sizes and different genders. More specifically, the study aimed to answer two research questions: (1) What can a temporal perspective tell us about the effect of table shape on student behaviour in different group sizes (dyads and triads) during a collaborative activity? and (2) What can a temporal perspective tell us about the effect of table shape on the behaviour of different genders (female and male))? To answer the research questions, the study focused on using ENA in order to better understand the effects of the learning space through modelling of co-occurrences of actions, given the limitations of traditional coding-and-counting approaches [16]. Our findings suggest that, in this data collection scenario, the two learning spaces affected triads and dyads, as well as female and male students, differently.

This study presents a different approach from previous ones adopted to date in the analysis of learning spaces. ENA has been employed in collaborative learning as well as in other areas [28,62], but the authors are not aware of this specific application in existing research on learning spaces. The findings on the co-occurrences of actions cannot be obtained using traditional coding-and-counting methods, which support the use of ENA in analysing learning spaces and contribute to the field of learning space design. The paragraphs below include discussion on the findings on the influence of table shape on students’ on-task actions during collaboration when considering two different group sizes and two different genders. Figure 7 is an overview of the ENA results, showing more prevalent co-occurrences of on-task actions in each case analysed.

### 5.1. Table Shape and Group Size

Starting with the first research question, the differences between conditions are visible both with dyads and triads, with a more pronounced difference in the case of dyads. Interestingly, the temporal analysis perspective yields different findings for these two group sizes. In dyads, students combine physical artefacts and non-verbal interaction more frequently when using round tables, as opposed to engaging in explanations and discussion while interacting with physical artefacts when working at rectangular tables. On the other hand, in triads, the round tables tend to foster more alternations between discussion and use of physical artefacts, and rectangular ones induce more co-occurrences of discussion and non-verbal interaction. While it cannot be said that the effects are explicitly opposed, it can be seen that the effect of more frequent co-occurrences between discussion and interaction with physical artefacts is caused by rectangular tables with dyads and by round tables with triads. Building on existing studies on the differences between dyads and triads, this study contributes by adding another explanatory factor that may shed light on previous findings on these differences and how the affordances of the space play a role in their behaviour [8,9]. The claim that triads are more likely to promote complex behaviours such as coalitions, negotiations, and conflict in discussions involving collaborative problem solving is more strongly substantiated in our scenario when round table shapes are used. The exchange of ideas and strategies that are shown to improve common performance also seem to be expressed differently in dyads in the two environments studied. The findings therefore support the hypothesis, based on the literature, that different table shapes can cause different student behaviour [42,45,68]. 

It should be observed that dyads and triads do not behave in the same way when the same table shape is used. The results, which show behavioural differences between dyads and triads, can expand previous research on this topic. Specifically, studies that deal with the differences between these two group sizes and report on the potential effect of the educational context [48,69] may be updated with the findings presented here. The physical context, indeed, is part of the educational context and should be considered when differences between dyads and triads are examined [69]. Therefore, by focusing on learning space and, more specifically, table shapes, in identifying differences between dyads and triads, this study extends previous research by contributing with new insights about group size. In terms of table shape, the findings indicate varying student behaviour according to group size. However, our results show that previous reports of the promotion of active discussion when round tables are used [70] can be confirmed only with triads. 

Furthermore, the findings that the use of physical artefacts, which co-occurs more frequently with some actions (discussion and non-verbal interaction) when both round and rectangular tables are used in dyads, as opposed to the same action’s co-occurrences with discussion in triads only when round tables are used, align with the literature [48]. That is to say, the hesitation that emerges in triads may have contributed to the reluctance of some group members to use artefacts. Furthermore, the literature establishes that dyads use equipment at a higher frequency when they are involved in practical work [47]. Additionally, a possible explanation for interaction with physical artefacts not being one of the actions that most often co-occur is the potential development of coalitions and conflicts, as stated in the literature [9], which leads to less use of artefacts by some students and dominance by others.

### 5.2. Table Shape and Gender

Considering the second research question and the effects of table shape on genders, the results suggest differences under varying conditions. The findings in the case of female students show more frequent co-occurrences of interaction with physical artefacts (IPA) and non-verbal interaction (Nv) when round tables are used. Conversely, male students at round tables exhibit more alternation between interaction with physical artefacts (IPA) and discussion (Ds). This behaviour of male students contrasts with the lack of verbal communication among female students under the same conditions, which indicates that the learning space exerts a different influence depending on gender. This difference may also be observed with rectangular tables, at which there are frequent co-occurrences between non-verbal interaction and off-task actions with female students, while more co-occurrences between non-verbal interaction and discussion occur with male students. Once again, the differing influences of table shape can be noted. 

The behaviour of female students confirms the findings of previous studies, in which the inequality between male and female group members engaged in engineering tasks like this one was evident [54]. Possible explanations could be the gender stereotype, which develops over years [10]. Furthermore, findings could be related to the potential existence of competitiveness, which has been shown to be demotivating for female participants in collaborative learning activity [14,15]. Observations detected a lower frequency of changes in the actions of female students while using certain artefacts, such as the Arduino, which may be attributed to the aforementioned uncertainty and the widespread belief that male team members possess greater knowledge. In contrast to male students, female students behave more passively but consistently during collaborative learning, which was also previously reported in the literature [51,52]. Although this study did not focus on group structure itself, it could be further extended by examining issues such as how gender distribution within groups affects collaboration, which has already been considered to some extent in the literature [50].

## 6. Limitations of the Study

The study has several limitations. One, common to studies in the field of educational technology in complex contexts [71], is the sample size. Organising studies with complex experimental setups in collaborative learning contexts, together with students’ potential time limitations and ethics (data sharing) concerns, places constraints on participant recruitment. Increasing the number of subjects in future studies will be especially important to further explore these questions from the perspective of gender, thereby leading to a greater understanding of gender differences with different group sizes and group composition. In the present study, a certain number of participants were removed from the analysis due to occlusion, which is a limitation in the analysis of collaborative activities with multiple participants when video recordings are used. More video cameras would help solve this problem in future studies. 

Another limitation is that the activity was specifically designed with a Jigsaw collaborative script that makes it difficult to generalise the findings to other collaborative learning designs. However, the activity was open-ended, which is common to a wide variety of collaborative activities. Furthermore, the jigsaw scrip allowed for the possibility to test dyads and triads during the same experiment in a structured way, while balancing each students’ load to provide the most similar conditions for each student as possible. Even so, collecting additional surveys from students on aspects such as the workload and emotional stress they experienced would be beneficial. Furthermore, pre- and post-surveys would be useful for understanding to what extent their behaviour was influenced by previously acquired experiences.

## 7. Conclusion and Future Research Lines

This study adds evidence to the fields of learning space design and collaborative learning by providing a setting for data collection and an analysis that enables observation of the interplay between table shape, group size, and gender, and their effects on on-task actions during collaboration. The results indicate the influence of table shape on student behaviour with different group sizes and different genders. Based on the ENA results, the different effects that different table shapes have on the course of student actions during collaboration have been identified. This study supports previous findings in the literature and extends them by providing further evidence that, due to its impact, the learning environment should not be overlooked as an important part of learning design. 

The temporal analysis perspective of collaborative behaviour adopted in this paper has been shown to be useful and meaningful in examining the varying effects of table shapes on different group sizes and genders. With this approach, the study contributes by applying known analysis methods in a new context. Namely, examination of temporal perspective has been shown to be meaningful in collaborative learning space analysis conducted in this study and a promising research direction for the future. Furthermore, the relationship between group size, gender and table shape, as a novel research perspective has the potential to further explore and contribute to research in collaborative learning fields. Regarding practical implications, more experimental research of table shape should be conducted in order to further clarify its role in the field of learning space research. In this way, the contribution will be made also in the field of learning design, by achieving adequate guidelines, especially when it comes to the size of groups and student behavior related to gender. 

## Figures and Tables

**Figure 1 sensors-21-02898-f001:**
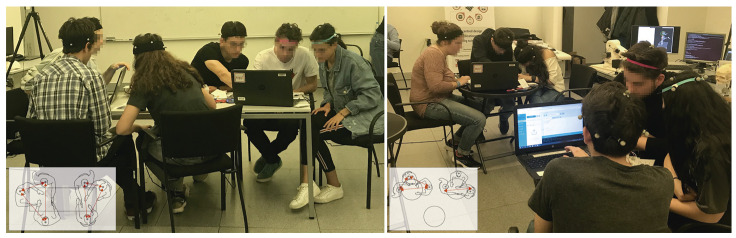
Two seating arrangements representing different conditions in the study.

**Figure 2 sensors-21-02898-f002:**
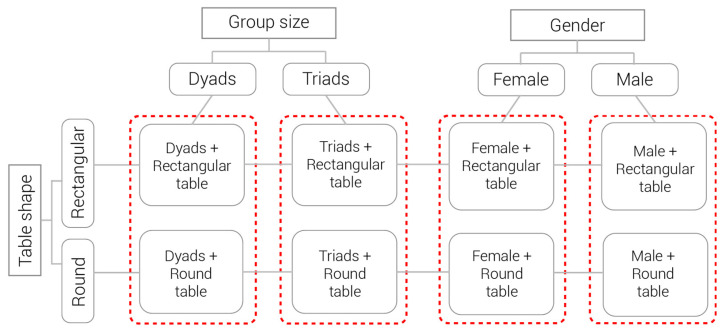
Four cases analysed.

**Figure 3 sensors-21-02898-f003:**
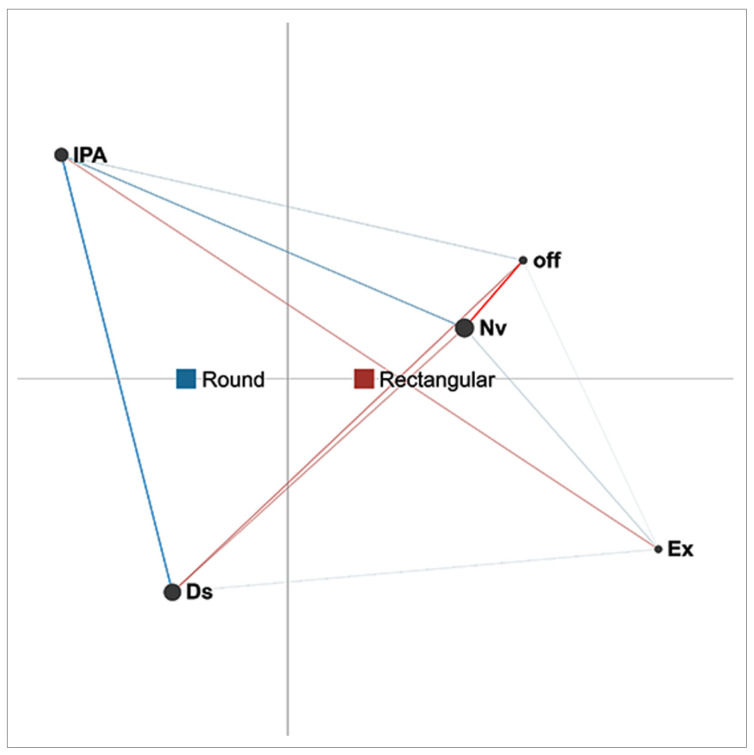
Difference network for triads under two conditions (round and rectangular tables).

**Figure 4 sensors-21-02898-f004:**
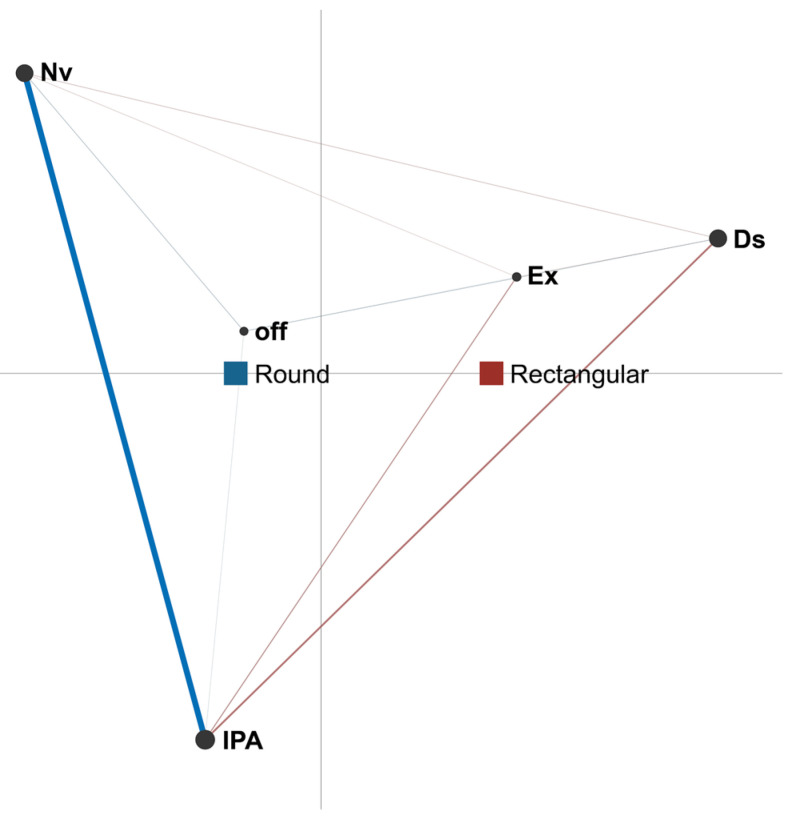
Difference network for dyads under two conditions (round and rectangular tables).

**Figure 5 sensors-21-02898-f005:**
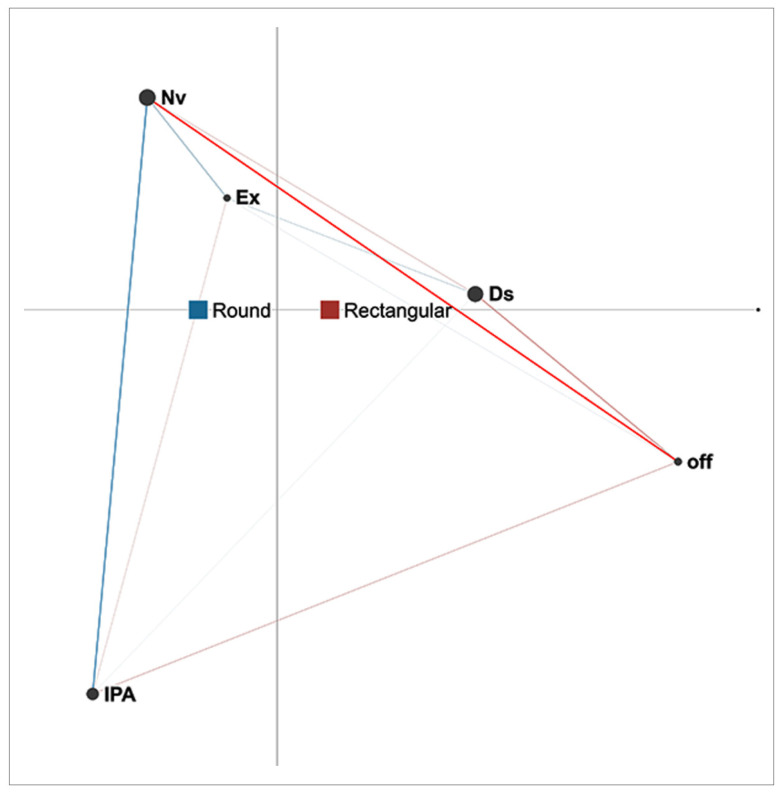
Difference network for female students under two conditions (round and rectangular tables).

**Figure 6 sensors-21-02898-f006:**
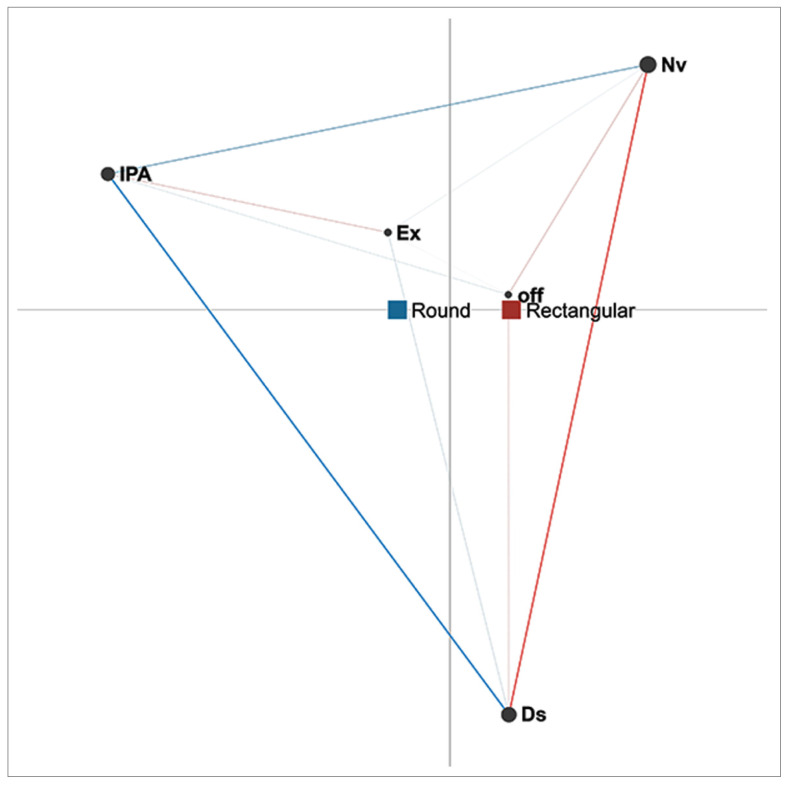
Difference network for male students under two conditions (round and rectangular tables).

**Figure 7 sensors-21-02898-f007:**
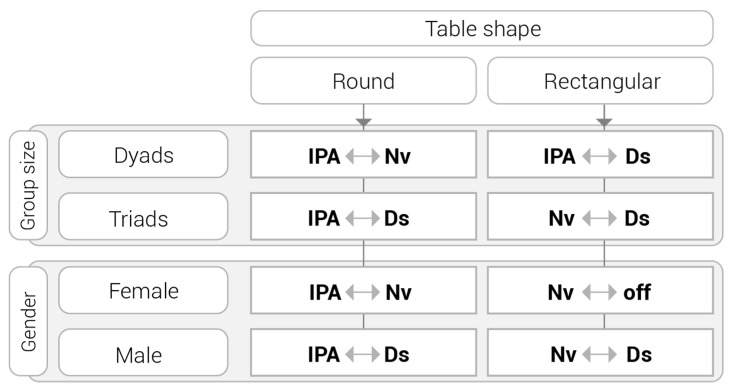
Summary of the epistemic network analysis (ENA) results of more prevalent co-occurrences of on-task actions for each case analysed.

**Table 1 sensors-21-02898-t001:** Coding schema for the analysis of student actions with abbreviations.

Code.	Explanation
Explanation (Ex)	Passive action (in terms of interaction)-the act or process of making something clear or easy to understand (telling, showing) without active participation from other participants. This is a social action that can overlap with physical actions (interaction with physical artefacts (IPA)).
Discussion (Ds)	Any type of discussion or quick exchange of words that includes interaction with participants (talking and pointing). This is a social action that can overlap with physical actions (interaction with physical artefacts).
Non-verbal interaction (Nv)	When a participant is not talking but is looking at teammates and/or gesturing as a sign of feedback (nodding, with ‘yes’ or ‘no’). This is a social action that can overlap with physical actions (interaction with physical artefacts).
Interaction with physical artefacts (IPA)	When participants use artefacts (Arduino, laptop, cards) to work individually or collectively. This physical action can overlap with social actions (*explanation*, *discussion*, and/or *non-verbal interaction*).
Off-task action (off)	Any action that is not directed towards the group, table or artefacts

**Table 2 sensors-21-02898-t002:** Examples of on-task actions.

Example of a Coded Action	Image Capturing Student Behaviour
Example 1 (group size-dyad): Both the student on the left side of the image and the student on the right are working with the artefacts (using instruction cards and writing ideas on the paper) without any verbal interaction. The action is coded in the following way: Student 1 (left side): non-verbal interaction (Nv), interaction with physical artefacts (IPA) Student 2 (right side): Nv, IPA	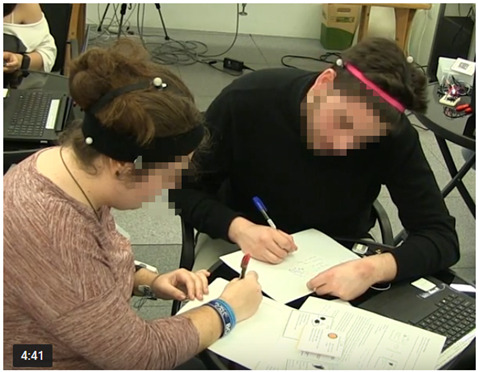
Example 2 (group size-dyad): The student on the left side of the image and the student on the right are talking to each other. They are not using artefacts and they exchange short sentences followed by words of agreement and nodding. The action is coded in the following way: Student 1 (left side): discussion (Ds) Student 2 (right side): Ds	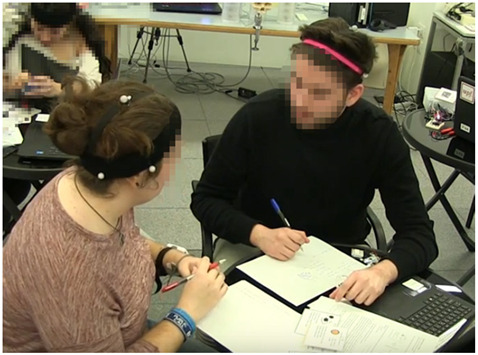
Example 3 (group size-triad): Student on the left side is presenting the idea while the student in the middle and student on the right look at him and the paper he is showing, and react verbally with head nodding. The action is coded in the following way: Student 1 (left side): explanation (Ex), IPA Student 2 (in the middle): Nv Student 3 (right side): Nv	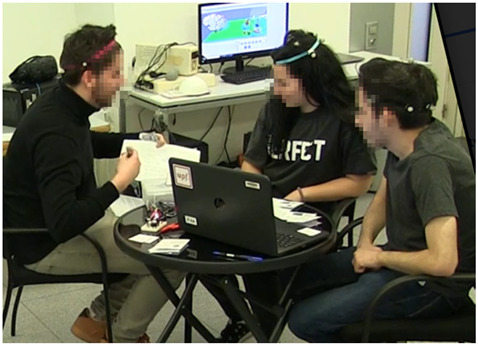
Example 4 (group size-triad) All three students are connecting elements and testing the Arduino system. The student on the left and the one in the middle are discussing something. The student on the right is also working with the Arduino system, but he is not talking. The action is coded in the following way: Student 1 (left side): Ds, IPA Student 2 (in the middle): Ds, IPA Student 3 (right side): Nv, IPA	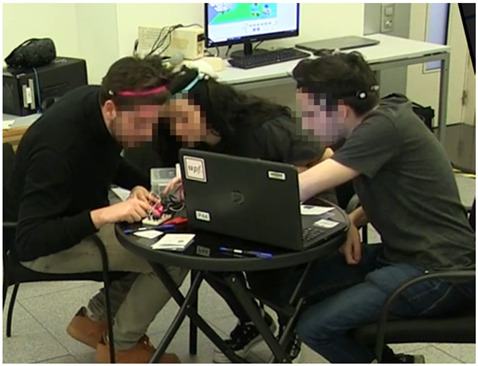

## Data Availability

Not applicable.

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
