# Peer review of "Studying Collaboration Dynamics in Physical Learning Spaces: Considering the Temporal Perspective through Epistemic Network Analysis"

_sensors, 2021, doi:10.3390/s21092898_

Round 1
Reviewer 1 Report
Dear authors,
Congratulations for your work. I think that you have done a good job.
I would suggest you only minor issues.
1) The gender differences that you have detected need more justification. Therefore I will suggest you see the paper titles below, which are related to gender differences:
a) Gender differences in kindergarteners’ robotics and programming achievement.
b) Latent Class Modeling of Children’s Preference Profiles on Tangible and Graphical Robot Programming
c) Age and gender differences in computer use and attitudes among secondary school students: What has changed?
d) Tangible and Graphical Programming with Experienced Children: A Mixed
Methods Analysis
e) Gender differences in older adults’ everyday cognitive collaboration
f) “Gender and computer games: Exploring females? dislikes
The papers above will be a good background and a good justification for the gender differences.
2) Please tell us a little more about the jigsaw method that at the 4.3 "Material and description section"
3) Please add more information at the 2.5 "Epistemic Network Analysis" section about the use of the method from other CSCL studies (eg. [54], etc.)
Thank you and regards,
Reviewer 2 Report
Review for „Studying Collaboration Dynamics in Physical Learning Spaces: considering the temporal perspective through Epistemic Network Analysis“
Summary
The paper presents the results of an evaluation if the adoption of the temporal perspective in the analysis offers insights into how different table shapes affect collaboration with different group sizes and different genders.
Bullet points
- Sentence on page two, line 67-70 is quite complex. Maybe rephrasing would be good
- Good introduction. Research gap is presented, but relevance for research including contributions for theory and practice should be fostered
- Good theoretical background on collaboration, learning spaces, temporality and ENA. However, it is unclear whether the authors cover “Group size and gender” in the theoretical background. The relevance for the study should be mentioned within the Introduction or a link has to be set within the background section “Indicators of fruitful collaborative learning”. Otherwise it seems a bit out of context (or random)
- I suggest moving the research aim and questions to the introduction. In addition the derivation seems quite random, especially considering the aspect of gender. “the paper also explores the gender perspective”. This seems like the authors found some evidence and did reverse engineering of this aspect. The same applies for group size. All aspects (group size, gender and table shape) need to be linked better within the introduction to form a consistent story line
- Good method, with rigor analysis. Very well reported including the use of pictures and other figures. This is important for the reader to understand the experimental setting.
- Good and comprehensive presentation of the results.
- Good discussion of the results. However, the implications for practice and especially for theory good be presented more. The limitations of the study are presented sufficiently.
Recommendation
Overall the paper presents a relevant topic (for the time after COVID) and analysis an ethnographic study in a rigor way. The paper is well-written and has a good theoretical background, method, discussion and results. However, I do find the derivation of the research questions quite generic and random. It seems that the researcher did reverse engineering after finding out that gender and group size might affect the collaboration. Maybe not, but both aspects are not properly derived in a consistent story within the introduction. Pointing out the relevance of both aspects and linking them better to collaboration and the table shape would enhance the paper.
I wish the authors good luck in continuing with their research.
